# Unusual Evolution of Hypertrophic Cardiomyopathy in Non-Compaction Myocardium in a Pompe Disease Patient

**DOI:** 10.3390/jcm12062365

**Published:** 2023-03-19

**Authors:** Vincenza Gragnaniello, Caterina Rizzardi, Anna Commone, Daniela Gueraldi, Evelina Maines, Leonardo Salviati, Giovanni Di Salvo, Alberto B. Burlina

**Affiliations:** 1Division of Inherited Metabolic Diseases, Department of Diagnostic Services, University Hospital, 35128 Padua, Italy; 2Division of Paediatric Cardiology, Department of Women’s and Children’s Health, University Hospital Padua, 35128 Padua, Italy; 3Division of Pediatrics, S. Chiara General Hospital, 38122 Trento, Italy; 4Clinical Genetics Unit, Department of Women’s and Children’s Health, and Myology Center, University of Padua, 35128 Padua, Italy

**Keywords:** Pompe disease, acid α-glucosidase, hypertrophic cardiomyopathy, non-compaction myocardium, cardiac remodeling, enzyme replacement therapy, alglucosidase alfa, cipaglucosidase alfa, miglustat, chaperone

## Abstract

Classic infantile Pompe disease is characterized by a severe phenotype with cardiomyopathy and hypotonia. Cardiomyopathy is generally hypertrophic and rapidly regresses after enzyme replacement therapy. In this report, for the first time, we describe a patient with infantile Pompe disease and hypertrophic cardiomyopathy that evolved into non-compaction myocardium after treatment. The male newborn had suffered since birth with hypertrophic cardiomyopathy and heart failure. He was treated with standard enzyme replacement therapy (ERT) (alglucosidase alfa) and several immunomodulation cycles due to the development of anti-ERT antibodies, without resolution of the hypertrophic cardiomyopathy. At the age of 2.5 years, he was treated with a new combination of ERT therapy (cipaglucosidase alfa) and a chaperone (miglustat) for compassionate use. After 1 year, the cardiac hypertrophy was resolved, but it evolved into non-compaction myocardium. Non-compaction cardiomyopathy is often considered to be a congenital, primitive cardiomyopathy, due to an arrest of compaction of the myocardium wall during the embryonal development. Several genetic causes have been identified. We first describe cardiac remodeling from hypertrophic cardiomyopathy to a non-compaction form in a patient with infantile Pompe disease treated with a new ERT. This has important implications both for the monitoring of Pompe disease patients and for the understanding of the pathophysiological basis of non-compaction myocardium.

## 1. Introduction

Classic infantile Pompe disease (IOPD) is the most severe form of Pompe disease (PD, OMIM #232300). PD is an autosomal recessive disease caused by the deficiency of the lysosomal enzyme acid α-glucosidase (GAA), responsible for the degradation of glycogen [1,2]. IOPD’s clinical phenotype is characterized by cardiac involvement (e.g., severe hypertrophic cardiomyopathy (HCM), arrhythmia), severe generalized hypotonia, and organomegaly [3]. Symptoms usually start during the first months of life. Without treatment, life expectancy is less than 1 year, with death often occurring due to cardiorespiratory failure [4]. The incidence of IOPD ranges from about 1/100,000 to 1/200,000, with variations in different populations [5]. 

Cardiomyopathy in IOPD is typically hypertrophic and can involve either the septum (asymmetric hypertrophy) or, more frequently, both the septum and the walls of the left and right heart (“concentric” hypertrophy). Both diastolic and systolic dysfunction can be observed [3]. 

The abnormal glycogen accumulation does not just affect the cardiomyocytes; conduction system cells can be affected as well, explaining the electrophysiological abnormalities often found in Pompe disease, such as pre-excitation patterns (short PR, delta waves), atrioventricular blocks, and bundle branch abnormalities [6].

The diagnosis of PD is established by demonstrating a deficiency in acid α-glucosidase enzyme activity in dried blood spots, lymphocytes, or fibroblasts, combined with finding disease-causing mutations of the *GAA* gene [7]. Urinary tetrasaccharide (uGlc4) and muscle necrosis enzymes (e.g., CPK, AST, ALT, LDH) are usually elevated. BNP can be elevated in patients with cardiac involvement [8]. To monitor cardiac involvement, two-dimensional echocardiography is the first-line imaging technique, as it is non-invasive, easily performed, and has a low cost. It allows evaluation of the left ventricular mass index (LVMI), septum and wall thickness, morphology, and function [9]. In some patients, cardiac magnetic resonance imaging (MRI) may be recommended to assess cardiac morphology. Gadolinium enhancement may be needed for cardiac tissue characterization [2]. Enzyme replacement therapy (ERT) is the only available treatment, and alglucosidase alfa is the only approved ERT. It is administered as an intravenous infusion every 1–2 weeks, at a dosage of 20/40 mg/kg [10]. The efficacy of the ERT may be influenced by the development of anti-ERT antibodies. The probability of developing high titers of anti-ERT antibodies primarily depends on patients’ cross-reactive immunological material (CRIM) status, as individuals who do not produce CRIM may develop more anti-ERT antibodies and may require protocols of preventive or therapeutic immunomodulation [11]. The bioavailability of the ERT is limited by its ability to bind with the mannose-6-phosphate receptor (M6PR) to enter muscle cells and lysosomes. New enzymes are being tested with higher affinity for M6PR, alone (e.g., avalglucosidase alfa) or in combination with a chaperone (e.g., cipaglucosidase + miglustat) [12,13,14].

ERT is usually effective in eliciting a significant regression of HCM in IOPD, which persists in long-term survivors [15]. Here, we describe the first case of HCM evolving into non-compaction myocardium (LVNC) after treatment in a patient with IOPD. 

## 2. Case Report

The patient—a male firstborn of unrelated parents of African descent (Burkina Faso)—was born at 37 weeks from an emergency delivery due to fetal heart failure. At birth, he was resuscitated and admitted to the NICU for invasive ventilatory assistance and circulation support. Echocardiography showed an HCM (LVMI 232 g/m^2^ Simpson’s method) with severe biventricular systolic and diastolic dysfunction (ejection fraction (EF) = 30%). The ECG showed a short PR interval (0.07 s). Moreover, the patient presented with severe hypotonia, along with elevated serum CPK (1063 U/L) and BNP (9169 U/L, nv 0-100). At 3 days of life, the extended newborn screening that has been mandatorily performed in our region since 2015 [16] showed positive results for PD (GAA activity in DBS 0.40 uM/L, nv > 3), and uGlc4 was elevated (71.2 mmol/mol creatinine, nv < 16.3). The diagnosis was confirmed by GAA molecular analysis (compound heterozygous c.2560C > T (p.Arg854*) + deletion exons 4-8), and the GAA Western blot assay on the lymphocytes showed a CRIM-negative status. After 5 days of life, the patient was treated with alglucosidase alfa (Myozyme^®^, Genzyme, Cambridge, MA, USA); simultaneously, an immunomodulation protocol with rituximab, EV immunoglobulins, and methotrexate was started [17]. In addition, an appropriate therapy for the heart failure was started (i.e., beta blockers, diuretics, ACE inhibitors). The patient progressively improved and, at the age of 2 months, he was discharged from the NICU. At 1 year, cardiac mass (LVMI 116.6 g/m^2^) and function (EF 50%) were improved, maintaining a compact aspect of the wall. At the same time, improvements in motor function and biomarkers (uGlc4 20.2 mmol/mol crea, nv <7.7, CPK 1877 u/L) were also recorded. Unfortunately, the patient subsequently developed a high anti-ERT antibody titer (1:102,400), followed by clinical and biochemical worsening (i.e., psychomotor delay, elevated serum CPK, maximum 6795 U/L, elevated uGlc4, maximum 50 mmol/mol creatinine). Cardiac mass was increased (LVMI 229 g/m^2^), as were BNP levels (237 U/L), while the EF remained stable (50%). The patient underwent a new immunomodulation cycle with bortezomib, rituximab, sirolimus, and EV immunoglobulins [18], with a reduction in the antibody titer to 1:6400 after 1 year, but with only partial clinical and biochemical improvements. In particular, he showed persistent psychomotor delay, HCM (LVMI 118 g/m^2^, EF 48%), increased uGlc4 (35 mmol/mol crea), and CPK (4728 U/L). At 2.5 years of life, he started a compassionate use therapy that combined a new ERT (cipaglucosidase 30 mg/kg/week) with a chaperone (miglustat 115 mg) (ATB200-15 Program, Amicus Therapeutics), with clinical benefit. After 3 months, the patient could walk independently, biomarkers were reduced (CPK 2989 U/L, uGlc4 32.6 mmol/mol crea), and cardiac mass (LVMI 82 g/m^2^) and function (EF 67%) improved, but echocardiography showed deep and large recesses and marked hypertrabeculation—especially in the apex. After 1 year from the start of the new therapy (3.5 years old), a new echocardiography assessment fulfilled the criteria of non-compaction myocardium (NC/C 2.6) with normal mass (LVMI 64 g/m^2^) and function (EF 75%). Cine MRI confirmed the diagnosis of biventricular non-compaction myocardium, evident in both the long-axis (Figure 1a) and short-axis views, with an NC/C ratio of 8.4/4.1 in medial segments of the left ventricle (Figure 1b), and with a maximum NC/C ratio of 11.4/2 in the apex (Figure 1c). The BNP titer was normal at the start (42 ng/L, nv 0-100) and throughout the course of the ERT/chaperone therapy. Disease biomarkers remained stable (CPK 2557 U/L, uGlc4 30.4 mmol/mol crea), and the patient had no cardiac symptoms. 

## 3. Discussion

In this report, we describe a case of cardiac remodeling from HCM to LVNC cardiomyopathy in a patient with IOPD treated with a new combined ERT/chaperone protocol (cipaglucosidase/miglustat). Diagnosis of LVNC was made by two-dimensional echocardiography, with the finding of a maximum ratio between the non-compacted and compacted myocardium greater than 2 at the end of the systole in the parasternal short axis, and evidence of numerous prominent trabeculations and ventricular cavity blood flow in deep intertrabecular recesses by color Doppler (according with Jenni criteria [19]) which was also confirmed by the MRI, with a ratio of the non-compacted to compacted myocardium of >2.3 in the diastole, according to the Peterson criteria [20]. 

Non-compaction cardiomyopathy is usually considered to be a primitive form of cardiomyopathy, rather than a secondary condition. In 2006, the American Heart Association classified this entity as primary cardiomyopathy of genetic origin [21].

This is a very rare form of congenital cardiomyopathy characterized by altered myocardial walls with two layers, consisting of a thin compacted layer of myocardium and a non-compacted myocardium (prominent trabeculae and deep intertrabecular recesses, filled with blood from the ventricular cavity) [22]. Non-compaction tissue is mostly found in the apex and in the lateral wall of the left ventricle, but in rare cases—such as our patient—it can affect the right ventricle or even both ventricles (22–38%) [23,24]. Although the precise cause is not known, LVNC is thought to be caused by an arrested compaction of the loose myocardial mesh during fetal ontogenesis [24]. During normal development, at around week 4 of human gestation, the linear heart tube begins to fold onto itself and to form prominent trabeculations, which contribute to the cardiac output, the nutrition of trabecular myocytes, and to oxygen uptake, before the development of coronary vascularization. During weeks 5 to 8, coronary vessels develop, and these trabeculae undergo a compression process. Disruption of this developmental process has been theorized to cause LVNC. Genetic mutations in the cytoskeletal proteins, mitochondrial function, sarcomeres, Z-lines, or actin and myosin components are regarded as possible etiologies, as they can affect the physiological development of the cardiac wall [25]. 

Considering that an increase in heart-wall trabeculations can be found both in physiological conditions (such as in athletes or pregnant women) and in pathologies such as arterial hypertension, neuromuscular disorders, hematological disorders, and kidney diseases, other authors hypothesize that acquired mechanisms can contribute to the pathogenesis of LVNC [26,27]. In particular, myocardial remodeling could be the effect of physiological adaptations after pressure overloads, or it could be a sign of myocardial damage in conditions characterized by contractile dysfunction, since an increase in the ventricular surface area may be an effort to compensate for reduced contractile force [28,29].

In conditions characterized by HCM, as in our patient’s case, the prominent trabeculae may be the result of an adaptation to greater demands for nutrition of the trabecular myocytes and an increased oxygen demand. 

In a study evaluating 211 HCM patients, an inverse correlation between wall thickness and trabeculation mass, as well as between EF and trabecular mass, was demonstrated [30]. 

Conversely, in our patient, trabeculations appeared after a rapid cardiac remodeling, associated with a great improvement in function (EF from 48% to 75%) and normalization of the LVMI (64 g/m^2^), with a shift from HCM to LVNC cardiomyopathy. We hypothesize that the new ERT/chaperone combination is more effective in reversing intracardial intralysosomal glycogen accumulation, leading to rapid improvement of the muscle function [31]. This process might result in unfavorable remodeling in some patients. Due to the rapid regression of ventricular hypertrophy experienced by our patient [15,32] and the necessity for the cardiovascular system to compensate for rapid changes, the LV trabeculae could become exposed and increase in number during the remodeling process [33], especially because our patient had previously shown an incomplete response to ERT. Very little is known about the cardiac outcome with the new protocol cipaglucosidase/miglustat, since our patient is only the second IOPD patient to receive this therapy. Continuous re-evaluation and longer follow-up are therefore necessary to better understand the impact that these drugs may have on cardiac function and remodeling, both in naive and in already-treated patients. To the best of our knowledge, this is the first reported case of PD shifting from HCM to LVNC. Moreover, although the coexistence of HCM and LVNC has been described, there are no descriptions of cardiac remodeling from a hypertrophic to a non-compaction cardiomyopathy after pharmacological treatment. This finding may shed new light on the physiopathology of LVNC and cardiac involvement in PD. 

## Figures and Tables

**Figure 1 jcm-12-02365-f001:**
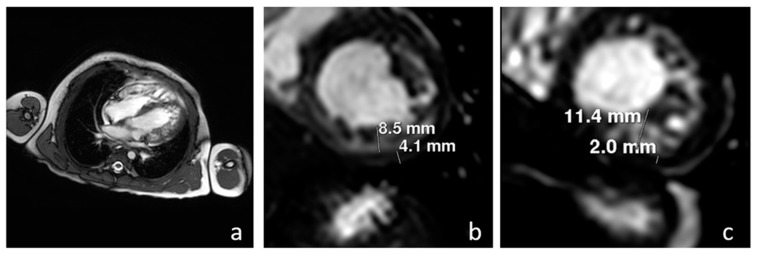
Cine MR performed after 12 months of therapy with the new protocol cipaglucosidase/miglustat, showing the presence of a biventricular non-compaction myocardium, evident in both the long-axis view (**a**) and the short-axis view, with an NC/C ratio of 8.4/4.1 in the medial segment of the left ventricle (**b**), and an NC/C ratio of 11.4/2 in the apex (**c**).

## Data Availability

Data available on request due to privacy/ethical restrictions.

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
