# Peer review of "Unusual Evolution of Hypertrophic Cardiomyopathy in Non-Compaction Myocardium in a Pompe Disease Patient"

_jcm, 2023, doi:10.3390/jcm12062365_

Round 1
Reviewer 1 Report
Dear editor and authors:
Thank you to invited me to review this manuscript. The authors presented a case of Pompe disease respond to treatment. I congratulate them on their successful experience in the treatment of this patient and appreciate their delicate observation in the treatment course. It is and interesting finding for readers. The manuscript was written in fluent English. The only question I would like to ask is about subsequent discovered LVNC cardiomyopathy. Is it possible that LVNC cardiomyopathy is congenital, but it was not discovered because the myocardium is too thick in infancy? If the LVNC cardiomyopathy is secondary. Is it a nature course of treatment effect or side effect of the drug.

Author Response
Reviewer 1
Dear editor and authors:
Thank you to invited me to review this manuscript. The authors presented a case of Pompe disease respond to treatment. I congratulate them on their successful experience in the treatment of this patient and appreciate their delicate observation in the treatment course. It is and interesting finding for readers. The manuscript was written in fluent English. The only question I would like to ask is about subsequent discovered LVNC cardiomyopathy. Is it possible that LVNC cardiomyopathy is congenital, but it was not discovered because the myocardium is too thick in infancy? If the LVNC cardiomyopathy is secondary. Is it a nature course of treatment effect or side effect of the drug.
R: We thank the reviewer for the comment and the important observation. LVNC cardiomyopathy is unlikely to be congenital because in the first 6 months of life cardiac mass improved with a compact aspect and the non-compactum aspect appeared only after the start of the new therapy with ciplaglucosidase/miglustat. It could be a secondary cardiomyopathy. Indeed, in conditions characterized by HCM, as in our patient’s case, the prominent trabeculae may be the result of an adaptation to greater demands of nutrition of the trabecular myocytes and an increased oxygen demand. However, in our patient trabeculations appeared after a rapid cardiac remodeling, associated with a great improvement of the function and normalization of the LVMI. Regarding the use of cipaglucosidase/miglustat, it is still experimental and our patient is only one of a few patients with classic infantile Pompe disease to be treated.

Reviewer 2 Report
The authors reported for the first time describe a patient with infantile Pompe disease and hypertrophic cardiomyopathy, that evolved in a cardiac remodeling to a non-compaction form after treatment with a new enzyme replacement/chaperon therapy. This has important implication both in monitoring of Pompe disease patients and in the understanding of the pathophysiological basis of non-compaction myocardium. Similar case has not been reported, therefore this report will give great information for the pediatric cardiologist and neonatologist. This is worthwhile.
To further strengthen the manuscript, following points should be clarified and addressed.
1. How was the BNP titer, decreased after ERT/chaperon therapy ?
2. Did you perform genetic analysis for LVNC?
Author Response
Reviewer 2
The authors reported for the first time describe a patient with infantile Pompe disease and hypertrophic cardiomyopathy, that evolved in a cardiac remodeling to a non-compaction form after treatment with a new enzyme replacement/chaperon therapy. This has important implication both in monitoring of Pompe disease patients and in the understanding of the pathophysiological basis of non-compaction myocardium. Similar case has not been reported, therefore this report will give great information for the pediatric cardiologist and neonatologist. This is worthwhile.
To further strengthen the manuscript, following points should be clarified and addressed.
- How was the BNP titer, decreased after ERT/chaperon therapy?
R: Thank you for your valuable comment. The BNP titer was normal at the start of therapy (42 ng/L, nv 0-100) and throughout the course of ERT/chaperon therapy. We have added this information in the manuscript.
- Did you perform genetic analysis for LVNC?
R: Thank you for the comment. The patient didn’t perform genetic analysis for LVNC. No family history for LVNC was reported and the parents declare they are not consanguineous. Moreover, before starting of the therapy with cipaglucosidase/migliustat, the patient did not show LVNC.
